# Vitamin D3 Supplementation at 5000 IU Daily for the Prevention of Influenza-like Illness in Healthcare Workers: A Pragmatic Randomized Clinical Trial

**DOI:** 10.3390/nu15010180

**Published:** 2022-12-30

**Authors:** Noud van Helmond, Tracy L. Brobyn, Patrick J. LaRiccia, Teresa Cafaro, Krystal Hunter, Satyajeet Roy, Brigid Bandomer, Kevin Q. Ng, Helen Goldstein, Ludmil V. Mitrev, Alan Tsai, Denise Thwing, Mary Ann Maag, Myung K. Chung

**Affiliations:** 1Department of Anesthesiology, Cooper University Health Care, Camden, NJ 08103, USA; 2Cooper Medical School of Rowan University, Camden, NJ 08103, USA; 3The Chung Institute of Integrative Medicine, Moorestown, NJ 08057, USA; 4Won Sook Chung Foundation, Moorestown, NJ 08057, USA; 5Rowan University School of Osteopathic Medicine, Stratford, NJ 08084, USA; 6Center for Clinical Epidemiology and Biostatistics Perelman School of Medicine, University of Pennsylvania, Philadelphia, PA 19104, USA; 7Cooper Research Institute, Cooper University Health Care, Camden, NJ 08103, USA; 8Division of General Internal Medicine, Cooper University Health Care, Camden, NJ 08103, USA; 9Division of Infectious Disease, Cooper University Health Care, Camden, NJ 08103, USA; 10Department of Family Medicine, Cooper University Health Care, Camden, NJ 08103, USA

**Keywords:** vitamin D, COVID-19, influenza-like illness, healthcare workers, hospital workers, randomized clinical trial

## Abstract

Vitamin D supplementation has been shown to reduce the incidence of acute respiratory infections in populations at risk. The COVID-19 pandemic has highlighted the importance of preventing viral infections in healthcare workers. The aim of this study was to assess the hypothesis that vitamin D3 supplementation at 5000 IU daily reduces influenza-like illness (ILI), including COVID-19, in healthcare workers. We conducted a prospective, controlled trial at a tertiary university hospital. A random group of healthcare workers was invited to receive 5000 IU daily vitamin D3 supplementation for nine months, while other random healthcare system workers served as controls. All healthcare workers were required to self-monitor and report to employee health for COVID-19 testing when experiencing symptoms of ILI. COVID-19 test results were retrieved. Incidence rates were compared between the vitamin D and control groups. Workers in the intervention group were included in the analysis if they completed at least 2 months of supplementation to ensure adequate vitamin D levels. The primary analysis compared the incidence rate of all ILI, while secondary analyses examined incidence rates of COVID-19 ILI and non-COVID-19 ILI. Between October 2020 and November 2021, 255 healthcare workers (age 47 ± 12 years, 199 women) completed at least two months of vitamin D3 supplementation. The control group consisted of 2827 workers. Vitamin D3 5000 IU supplementation was associated with a lower risk of ILI (incidence rate difference: −1.7 × 10^−4^/person-day, 95%-CI: −3.0 × 10^−4^ to −3.3 × 10^−5^/person-day, *p* = 0.015) and a lower incidence rate for non-COVID-19 ILI (incidence rate difference: −1.3 × 10^−4^/person-day, 95%-CI −2.5 × 10^−4^ to −7.1 × 10^−6^/person-day, *p* = 0.038). COVID-19 ILI incidence was not statistically different (incidence rate difference: −4.2 × 10^−5^/person-day, 95%-CI: −10.0 × 10^−5^ to 1.5 × 10^−5^/person-day, *p* = 0.152). Daily supplementation with 5000 IU vitamin D3 reduces influenza-like illness in healthcare workers.

## 1. Introduction

Healthcare workers are frequently affected by acute respiratory tract infections [1] and have a higher risk of acquiring coronavirus disease 2019 (COVID-19) than the general population [2]. In addition to the individual disease burden to sick workers, the potential pathogen transmission by sick workers creates a serious patient safety concern [3,4]. Furthermore, illness among hospital workers has often created staffing shortages in vital positions during the current COVID-19 pandemic [5].

Outside the COVID-19-specific literature, meta-analyses have found that vitamin D supplementation taken over weeks to months reduces acute respiratory infections [6,7,8]. This effect is more pronounced in individuals with the lowest baseline serum concentrations of vitamin D but remains significantly independent of baseline levels [6,7,8]. Importantly, the protective effect is more pronounced when vitamin D is supplemented in daily doses, when compared to large boluses [6,7,8]. Severe acute respiratory syndrome coronavirus 2 (SARS-CoV-2) enters human cells through cell surface angiotensin-converting enzyme 2 receptors [9]. Inadequate blood concentrations of vitamin D are associated with inappropriate activation of the renin-angiotensin-aldosterone system [10]. This mechanistic link between vitamin D concentrations and COVID-19′s pathophysiology, combined with the body of evidence supporting vitamin D supplementation as a method to reduce other respiratory tract infections, led to calls for prospective clinical studies on vitamin D supplementation for COVID-19 [11].

Early in the COVID-19 pandemic, prospective clinical trials on vitamin D supplementation for COVID-19 assessed if vitamin D could ameliorate the clinical course in patients with various levels of COVID-19 severity. Repeated high-dose vitamin D supplementation improved viral clearance in asymptomatic or mildly symptomatic patients [12], as well as reduced the need for intensive care unit treatment in hospitalized patients [13]. However, a single high dose of vitamin D did not reduce hospital length of stay in hospitalized patients with moderate to severe COVID-19 [14]. While the pre-COVID evidence for an effect of vitamin D on respiratory tract infection is found predominantly in preventing acute disease [6,7,8], prospective studies on vitamin D supplementation for COVID-19 prevention have been absent. Two recent studies by Jolliffe et al. and Vallasis et al. had conflicting results [15,16].

Randomized controlled trial (RCT) literature on vitamin D for the prevention and management of respiratory infection is more extensive than RCT literature regarding vitamins A and C. The aforementioned has summarized significant literature available on vitamin D in the prevention of respiratory infection which includes a meta-analysis of 43 RCTs [6]. A meta-analysis of 9 RCTs studying the effect of vitamin C on the common cold found inconsistent results t with no attention to its prophylactic effects [17], whereas a systematic review of 40 RCTs on vitamin A found a lack of effect regarding influenza virus and coronavirus prevention and management [18].

The aims of this study were to assess if daily vitamin D can prevent non-COVID-19 and COVID-19-related influenza-like illness (ILI) in healthcare workers. During the COVID-19 pandemic, healthcare workers in our hospital were required to self-monitor for symptoms of ILI and report to our employee health service for COVID-19 testing. We hypothesized that daily vitamin D3 supplementation in healthcare workers leads to reduced incidence of ILI due to reduction in both non-COVID-19 ILI and COVID-19 ILI when compared to other healthcare workers.

## 2. Materials and Methods

### 2.1. Subjects and Regulatory Approval

Starting in October 2020, and over the course of 13 months, we conducted a prospective study (Clinicaltrials.gov: NCT04596657) on healthcare workers to investigate two aims: 1. To assess if daily vitamin D3 can prevent ILI in healthcare workers; 2. To assess if daily vitamin D can prevent COVID-19 in healthcare workers. A randomly selected group of healthcare system employees were approached for participation in the study to take daily vitamin D3 supplementation for nine months. Workers aged 18 years or older were eligible to participate; exclusion criteria consisted of conditions or medications and supplements that could increase health risk by receiving vitamin D supplementation (Table 1). All participants provided written informed consent after all study information was provided in detail and an opportunity to ask questions was provided. The local Institutional Review Board approved this study (IRB #20-455). This study adhered to CONSORT guidelines [19]. The CONSORT checklist is included in Appendix A.

### 2.2. Recruitment and Randomization

Using a list of all healthcare system workers aged 18 years or older, we randomized workers to the intervention (vitamin D3) and control groups (Figure 1). We used Zelen’s design [20,21,22,23,24] for this study, which is a pragmatic clinical trial design whereby subjects are randomized prior to informed consent, and wherein initially only subjects randomized to the interventional arm are approached for consent and subsequently enrolled. Our initial intention was to enroll only subjects aged 52 and older with 1:1 randomization between the groups. The study was conducted during the winter period in which acute respiratory infections are known to spike. A power calculation was based on the rate of presenting with influenza-like illness during the spring and summer of the same year, to provide a conservative estimate. Extrapolating this rate to a 9-month study duration, it was expected that 34.4% of the control group would report to Employee Health for at least one respiratory infection. A detectable reduction of 6% in at least one acute respiratory infection due to vitamin D3 supplementation was assumed. With a one-tailed α of 0.05 and β of 0.15, 859 subjects were projected in the vitamin D supplementation and control arms. Early in the trial, it became apparent that recruitment was low, thus the lower limit of age for eligibility to participate in the trial was changed from 52 to 18.

Randomization was performed by the biostatistician and a co-investigator utilizing the RAND function in Microsoft Excel. When the younger adults were allowed to be enrolled in the trial, we switched from 1:1 to 2:1 randomization (intervention:control). Subjects randomized to the intervention arm were contacted by email to assess their interest in receiving daily vitamin D3 supplementation. Those randomized employees interested in participating were screened for inclusion and exclusion criteria. Those in the passive control group were contacted towards the end of the study to voluntarily complete a survey that included their informed consent, medical history, vitamin D intake history, and COVID-19 vaccination history. The pragmatic design of the study was the basis for contacting control subjects toward the end of the study. Zelen’s design is ethical and particularly useful within the context of trials of prevention and screening interventions [21,22,26,27]. A flow diagram outlining the randomization and consent procedures is presented in Figure 1.

### 2.3. Vitamin D

Vitamin D3 was supplied to eligible participants in oral gel capsules at a dose of 5000 IU per day (Res-Q Vital D3, N3 Oceanic Inc, Pennsburg, Pennsylvania, USA). A daily dose of 5000 IU is required to attain normal serum 25(OH)D concentrations in individuals who have concentrations below 55 nmol/L at baseline without supplementation [25]. Furthermore, in the state of New Jersey where this study was conducted, 28% of adults over the age of 20 are obese [28], and obese individuals require 2–3 times the normal dose of vitamin D supplementation for vitamin D deficiency [29]. The protective effect of vitamin D supplementation on acute respiratory tract infections that was found in systematic reviews in individuals without particularly low serum concentrations of 25(OH)D supports providing supplementation of vitamin D3 to individuals who may not be deficient in serum vitamin D by current clinical standards [6,7,8].

Pillboxes were provided to promote adherence to vitamin D3 supplementation (Pill Thing, Inc., Ellisville, MO, USA). Biweekly automated text and email reminders (Twilio, San Francisco, CA, USA) were used as an additional method to support adherence to the study supplementation. Emails included a picture of the vitamin D3 supplement. Vitamin D3 supplementation was provided for three months at the start of the study, and again at three and six months of study participation.

### 2.4. Survey Data Collection

Using an electronic data capturing system (REDCap [30,31], Vanderbilt University, Nashville, TN, USA), we collected information on demographic characteristics, medical history, vitamin D deficiency history, COVID-19 history, as well as influenza and COVID-19 vaccination status. In the intervention group, we additionally collected information on self-reported adherence to the daily vitamin D3 study supplementation. Adherence was monitored in two ways: 1. Monthly surveys asking subjects to report the number of missed doses and 2. Every three months over the 9-month study period subjects were asked on their monthly survey to count the number of pills left in their bottle which originally contained a 3-month supply.

### 2.5. Primary and Secondary Outcome Measurement

As part of standard employee health and patient safety procedures during the COVID-19 pandemic, all healthcare system workers were required to self-monitor daily for ILI symptoms. Symptomatic workers were required to contact employee health services. The employee health department then prescribed swab polymerase chain reaction (PCR) testing for these employees considering the possibility that these symptoms were due to COVID-19. Our primary outcome, the incidence rate of ILI, was defined as workers who were referred for testing at least once during the study period. The secondary outcomes, the incidence rate of COVID-19 ILI and non-COVID-19 ILI, were defined as at least 1 positive and at least 1 negative COVID-19 PCR test(s) during the study period, respectively.

### 2.6. Safety Assessments

Considering the excellent safety profile of vitamin D3 at a dose of 5000 IU/day [32,33,34,35,36], we did not include laboratory testing or other clinical interventions in our procedures unless clinically indicated. Subjects were monitored via monthly surveys that queried subjects on symptoms of hypercalcemia and nephrolithiasis which included 15 symptoms [37]. In addition, subjects were asked if they had experienced any new or unusual health changes and if they were taking any new medications. If subjects indicated symptoms, health changes, or new medications, additional information was collected through open-ended questions and they were contacted by a study investigator by phone to obtain more information. Symptoms of concern were shared with the primary safety monitor (SR) who ordered laboratory tests; scheduled outpatient visits for evaluation; or referred the subject for additional evaluation when indicated. The safety population included all individuals in the intervention group. Adverse events (AEs) were defined as treatment-emergent AEs if they occurred on or after the date of the first dose of vitamin D3 and up to the end of each subject’s study period.

### 2.7. Adherence

Supplementation adherence rates were calculated by dividing the number of pills taken by the number of pills that should have been taken. During the study, subjects reported the number of pills remaining in bottles 1, 2 and 3 at survey months 3, 6 and 9, respectively. The total adherence calculation for subjects with complete adherence data is the sum of pills taken at months 3, 6 and 9 (days 90, 180 and 270) divided by 270 (i.e., sum of pills in bottles 1, 2 and 3). Withdrawn subjects were included, as well as subjects who had periods of vitamin D3 interruption. In these cases, the denominator was adjusted accordingly. Any subjects with missing adherence data (i.e., missing pill counts at months 3, 6 or 9) or incongruent data were not included in the adherence calculation.

### 2.8. Data & Statistical Analysis

Characteristics of the intervention and control group are presented as mean ± standard deviation or *n* (%). Workers in the intervention group were included in the analysis if they completed at least 2 months of supplementation, to allow for adequate vitamin D plasma levels [25]. Workers who discontinued supplementation after at least 2 months of supplementation were included up to the time of discontinuation or termination of employment. To provide an objective means to identify meaningful differences in demographic and clinical characteristics between the intervention and control groups we used standardized mean differences with a cutoff of 20% or 0.20 [38,39,40], see Table 2. As our resulting sample size was suboptimal (70.3% short of the targeted sample size), we conducted a per protocol analysis; an intention-to-treat analysis would require all of the subjects who did not elect to participate to be calculated as part of the intervention group. To achieve the study aims, we compared the incidence rate of all ILI determined by PCR testing for COVID-19 conducted by employee health in the intervention and control group subjects. The incidence rate was expressed as workers who had experienced at least 1 ILI (positive or negative SARS-CoV-2 PCR test results) per person-day. Incidence rates were used, rather than incidence, to adjust for the lengths of the observation periods. Separate analyses were performed for the incidence rate of COVID-19 ILI (positive SARS-CoV-2 PCR test results) and non-COVID-19 ILI (negative SARS-CoV-2 PCR test results). Confidence intervals of the difference in rates were calculated using the “Test Base Method” and *p*-values were obtained using the Chi-square statistic [41]. The fact that no cases of certain predefined endpoints occurred in a study arm precluded the use of Poisson and negative binomial regression models. To assess the influence of COVID-19 vaccination, secondary analyses were performed on incidence rates including only the observation period after a worker was fully vaccinated (14 days after the second dose of an mRNA vaccine [42,43] or after the first dose of the non-mRNA Ad26.COV2.S vaccine [44]). MedCalc (version 20, MedCalc Software Ltd., Ostend, Belgium) was used for all statistical analyses and Prism (version 9, GraphPad Software Inc, San Diego, CA, USA) was used to create graphs.

## 3. Results

### 3.1. Trial Enrollment and Baseline Characteristics

Of our targeted sample size of 859 subjects, 299 healthcare system workers were enrolled between 27 October 2020 and 31 January 2021 to participate in the intervention group. A total of 255 subjects completed vitamin D3 supplementation for at least 2 months and were included in the analyses. The last study participant completed nine months of vitamin D3 supplementation on 23 November 2021. During the same period, 2892 random healthcare system employees were passively enrolled to constitute the control group. Five hundred and seventy-eight control group participants provided demographic and clinical information at the completion of the study period. Detailed trial flow is presented in the CONSORT trial flowchart—Figure 1. Demographic and clinical characteristics were similar between the vitamin D3 supplementation intervention and control groups—Table 2. The average age was slightly higher in the control group (50 vs. 47) years. The average adherence rate of vitamin D3 supplementation was 87% with a median rate of 91%.

The total observation time in the intervention and control groups was 49,147 and 861,141 person-days, respectively. The not-fully vaccinated observation periods comprised 5879 and 240,784 person-days in the intervention and control groups, respectively, whereas fully vaccinated observation periods were 43,268 and 620,357 person-days.

### 3.2. All Influenza-like Illness

In the intervention group, three workers had at least 1 episode of ILI, while 197 workers in the control group had at least 1 episode of ILI. Incidence rates of ILI in the vitamin D3 supplementation and control groups are presented in Table 3 and Figure 2. Comparison of incidence rates revealed that vitamin D3 supplementation was associated with a lower risk of ILI (absolute incidence rate difference −1.7 × 10^−4^/person-day, 95%-CI −3.0 × 10^−4^ to −3.3 × 10^−5^/person-day, *p* = 0.0147). Absolute incidence rate differences were −4.653 × 10^−4^/person-day, 95%-CI −1.1118 × 10^−3^ to 1.811 × 10^−4^/person-day, *p* = 0.1583 and −2.632 × 10^−5^/person-day, 95%-CI −1.0833 × 10^−4^ to 5.57 × 10^−5^/person-day, *p* = 0.5294 when only considering non-fully vaccinated and fully vaccinated observation periods in participants, respectively—Table 4 and Figure 3 and Figure 4.

### 3.3. COVID-19 Influenza-like Illness

In the intervention group, no workers had a positive COVID-19 PCR test during the observation period, while in the control group 36 workers had at least 1 positive COVID-19 PCR test. Incidence rates in the vitamin D3 supplementation and control groups are presented in Table 3 and Figure 2. A comparison of incidence rates revealed that vitamin D3 supplementation was associated with a non-statistically significant lower risk of COVID-19 ILI (absolute incidence rate difference—4.181 × 10^−5^/person-day, 95%-CI −9.897 × 10^−5^ to 1.536 × 10^−5^/person-day, *p* = 0.1517). The absolute incidence rate difference was −1.495 × 10^−4^/person-day, 95%-CI −4.621 × 10^−4^ to 1.63 × 10^−4^/person-day, *p* = 0.3485 when only considering non-fully vaccinated observation periods in participants; no COVID ILI occurred in either group during the fully vaccinated observation period—Table 4 and Figure 3 and Figure 4.

### 3.4. Non-COVID-19 Influenza-like Illness

In the intervention group, three workers had at least 1 episode of non-COVID-ILI, while 165 workers in the control group had at least 1 episode of non-COVID-ILI. Incidence rates of non-COVID ILI in the vitamin D supplementation and control groups are presented in Table 3 and Figure 2. A comparison of incidence rates revealed that vitamin D supplementation was associated with a lower risk of non-COVID-ILI (absolute incidence rate difference −1.306 × 10^−4^/person-day, 95%-CI −2.541 × 10^−4^ to −7.1 × 10^−6^/person-day, *p* = 0.0382). Absolute incidence rate differences were −3.324 × 10^−4^/person-day, 95%-CI −9.078 × 10^−4^ to 2.43 × 10^−4^/person-day, *p* = 0.2575 and −2.632 × 10^−5^/person-day, 95%-CI −1.0833 × 10^−4^ to 5.57 × 10^−5^/person-day, *p* = 0.5294 when only considering non-fully vaccinated and fully vaccinated observation periods in participants—Table 4 and Figure 3 and Figure 4.

### 3.5. Treatment-Emergent Adverse Events (TEAEs)

Among the 299 enrolled subjects, 182 workers in the intervention group reported at least 1 TEAE(s). Out of 388 reported TEAEs, 80% were judged to be unrelated, 18% possibly related, and 2.1% probably related. See Appendix A for a comprehensive listing of all TEAEs. There was one unrelated serious adverse event: hospitalization due to ruptured Meckel’s diverticulitis.

The 8 probably related TEAEs occurred in five participants. One participant reported 3 adverse events that began after 7 months on study vitamin D3: burning sensation in arms and legs; oral tenderness; and hypersensitivity to the sun. Interruption of vitamin D3 for 15 days showed a slight improvement in symptoms. Upon resuming vitamin D3 for two weeks, the subject reported worsening symptoms. Abatement of all symptoms was reported approximately one month after the final vitamin D3 discontinuation. The subject was withdrawn from the study. One subject reported two adverse events after 5 weeks of vitamin D3: thirst; and frequent urination. The symptoms abated after discontinuing vitamin D3. Upon resumption, symptoms returned. The subject was withdrawn from the study. One subject reported worsening back and right leg pain. The subject discontinued vitamin D3 after approximately 5 weeks of study due to worsening pain. Within one to two weeks of discontinuation, the subject reported that pain levels returned to baseline. The subject was withdrawn from the study. One subject was on vitamin D3 for approximately two months when she developed mouth lesions. The subject stopped vitamin D3 for 1 week and the lesions resolved. Upon resuming vitamin D3 for a few days, a small lesion appeared. The subject stopped vitamin D3 and discontinued involvement with the study. One subject reported mild abdominal cramping after 10 days on vitamin D3. The subject stated the issue was resolved after changing the time of day (not provided) she took vitamin D3, and the subject completed the study.

Three in-person clinic visits occurred to examine subjects for palpitations, joint pain/depression/nausea and lower back pain/leg pain. Laboratory tests (vitamin D, calcium, phosphorus, magnesium levels) were ordered for a total of 22 subjects. There were no instances of hypercalcemia reported. One subject reported nephrolithiasis on a survey 1 month after the event and was immediately withdrawn. The subject’s calcium level was within normal limits and the nephrolithiasis was considered unrelated to the study vitamin D3. Two subjects were withdrawn due to pregnancy.

Consistent with our protocol’s provision for privacy, adverse events were not collected in the passive control group.

### 3.6. Crossover

Five hundred seventy-eight (20%) control group subjects responded to the near-end-of-study survey that included questions regarding vitamin D3 intake: 267 (46.2%) of the respondents did not take any vitamin D3; and 311 (53.8%) did take vitamin D3. Of the 311 subjects, 205 provided sufficient information. The mean dose of those 205 respondents was 3034 IU with a median dose of 2000 IU. For all respondents providing sufficient information (472) the mean dose was 1318 IU with a median dose of 0.

## 4. Discussion

### 4.1. Principal Findings

The aim of this study was to assess the hypothesis that daily vitamin D3 supplementation reduces the incidence of ILI in healthcare workers. We found that daily vitamin D3 supplementation was associated with overall reduced ILI. Contributions to this outcome came from the differences in non-COVID-ILI incidence rates throughout the entire study period and COVID-19 ILI incidence rates during the period in which healthcare workers were not vaccinated for COVID-19. Overall, a low proportion of healthcare workers tested positive on PCR testing for COVID-19 during the study.

### 4.2. Influenza-like Illness

Our primary finding is consistent with recent large meta-analyses that have looked at the effect of vitamin D3 supplementation on the incidence of respiratory tract infections [6,7,8]. Joliffe et al. performed a systematic review and meta-analysis of 43 double-blinded randomized controlled trials on the protective effect of vitamin D3 supplementation on respiratory tract infections and found that supplementation reduced the risk compared to placebo with an odds ratio (OR) of 0.92 (95% CI 0.86–0.99) [6]. They found that daily supplementation was particularly protective when they separately analyzed the studies that provided daily dosing regimens in contrast to intermittent or single-dose regimens (OR 0.78, 95% CI 0.65–0.94).

### 4.3. COVID-19 ILI

The secondary aim of this study was to assess if daily vitamin D3 supplementation could reduce ILI due to COVID-19 in healthcare workers. We found that compared to the control group, the vitamin D3 supplementation group experienced a non-statistically significant lower rate of ILI due to COVID-19. Several early pandemic prospective therapeutic studies have explored the effect of vitamin D supplementation on disease course in COVID-19 patients. Repeated high-dose vitamin D supplementation improved viral clearance in asymptomatic or mildly symptomatic patients [12]. Repeated administration of vitamin D to hospitalized patients with COVID-19 reduced the need for intensive care unit treatment in hospitalized patients [13]. However, a randomized controlled trial that assessed if a single high dose of vitamin D3 could reduce the hospital length of stay in patients with moderate to severe COVID-19, found no effect of vitamin D compared to placebo [14]. Orally provided vitamin D2 or D3 needs to be hepatically converted to 1,25-dihydroxyvitamin D for biological activity. It is highly questionable if critically ill patients are able to efficiently convert vitamin D delivered in a single very large bolus dose, to active 1,25-dihydroxyvitamin D [45].

More recently, a randomized controlled trial on disease prevention with vitamin D supplementation of 800 IU or 3200 IU per day for 6 months in 6200 participants found no difference in either acute respiratory tract infections or in COVID-19 incidence [15]. In contrast, our study found a clear difference in any influenza-like illness (ILI) incidence and a non-statistically significant reduction in COVID-19 incidence. Our study employed a larger dose of vitamin D3 (5000 IU versus 3200 IU). In addition, our study ensured adequate vitamin D levels by requiring 2 months of vitamin D3 intake, while Jollife et al. obtained levels at the end of their study, leaving unknown the levels during most of the observation period [15]. Villasis-Keever et al. demonstrated vitamin D to be protective against COVID-19 at a dose of 4000 IU [16].

### 4.4. Clinical Implications

Daily vitamin D3 supplementation is an inexpensive and safe intervention [36]. The results of the present study suggest that daily vitamin D3 supplementation can be recommended to reduce the incidence of ILI in healthcare workers. Reducing ILI is important to improve the health of healthcare workers [1], to improve patient safety due to less exposure to sick healthcare workers [3,46], and to reduce health care utilization and health care cost [47,48]. Our result of a COVID-19 incidence reduction that was not statistically significant points to the necessity for a sufficiently powered clinical trial. Furthermore, and in the context of the vaccinated population, it appears a large sample size will be required to determine if vitamin D supplementation can help prevent breakthrough COVID-19 infections.

### 4.5. Methodological Considerations

Several methodological considerations pertain to the present study. First, we used a Zelen randomized study design in which individuals were selected to be eligible for active study participation. From the group of selected individuals, a proportion of individuals ultimately decided to participate, which introduces the possibility of selection bias. However, we stress that the enrollment was not outcome-dependent and therefore should not bias our results unless the prognosis in the intervention group differed from the control group. We compared intervention group patients to control group patients and found no significant differences between groups for a range of demographic and clinical characteristics, except for age which was slightly above the predefined standardized difference threshold (standardized difference 0.24). Additionally, we did not report influenza vaccination information on an individual participant level due to ambiguity in survey response data. However, because our study was conducted in a healthcare setting, and influenza vaccination was mandatory at the time, there was a very high overall influenza vaccination rate (86%) when we started to count influenza-like illness incidence, which was in January 2021. Despite the possibility that flu vaccination could have biased the study toward the null, our results indicated a positive and statistically significant difference. In conclusion, we have no indication that enrollment biased results. Additionally, we used an unblinded pragmatic design, which could have introduced observer bias.

## 5. Conclusions

Daily supplementation with 5000 IU vitamin D3 can be used to reduce influenza-like illness in healthcare workers. A sufficiently powered study to assess the reduction in COVID-19 breakthrough infections is recommended.

## Figures and Tables

**Figure 1 nutrients-15-00180-f001:**
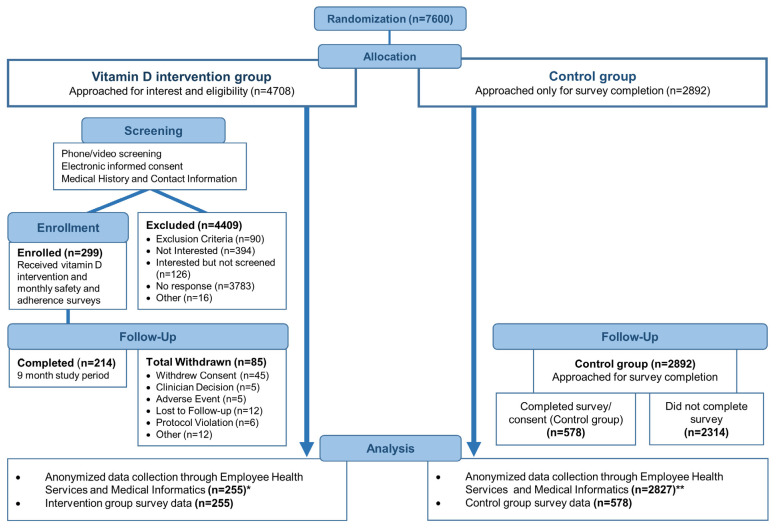
CONSORT trial flowchart. * Out of 299 intervention subjects, 44 subjects were excluded from analysis because they were not on 5000 IU vitamin D3 per day for at least 60 days, a timeframe that is considered to be protective [25]. The resulting 255 subjects included subjects who completed the 9-month study period and early withdrawals. ** Employee Health was unable to locate 27 control subject records for unknown reasons. A possibility for missing records could be due to name changes over the course of the study. Another 38 control subjects were found to have been terminated from employment prior to the study observation period and were excluded from analyses (*n* = 2892 − 27 − 38 = 2827).

**Figure 2 nutrients-15-00180-f002:**
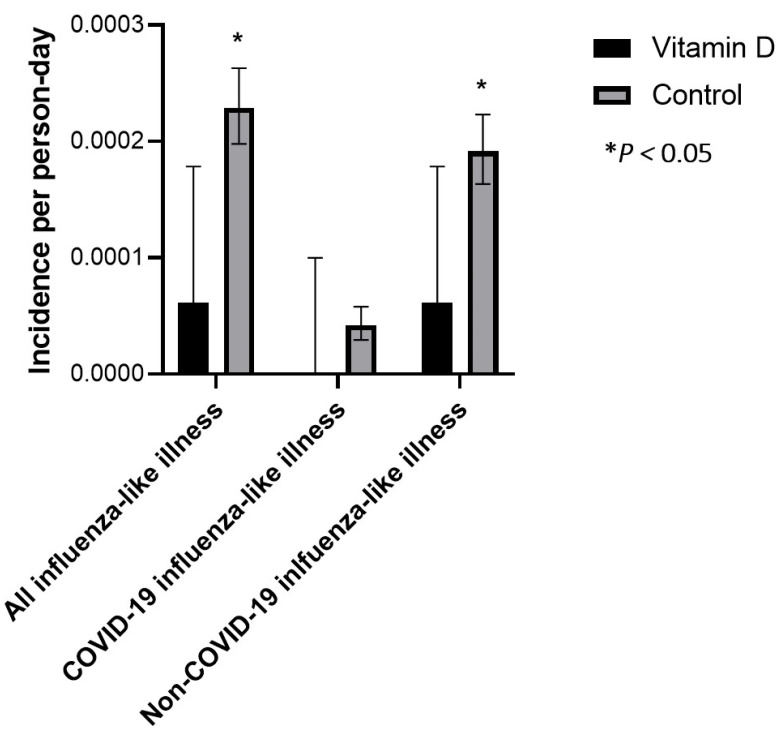
Incidence rate of influenza-like illness, COVID-19, and non-COVID-19 influenza-like illness in the vitamin D supplementation and control groups. 95% confidence intervals are based on the methods described by Sahai and Khurshid [41]. * *p <* 0.05.

**Figure 3 nutrients-15-00180-f003:**
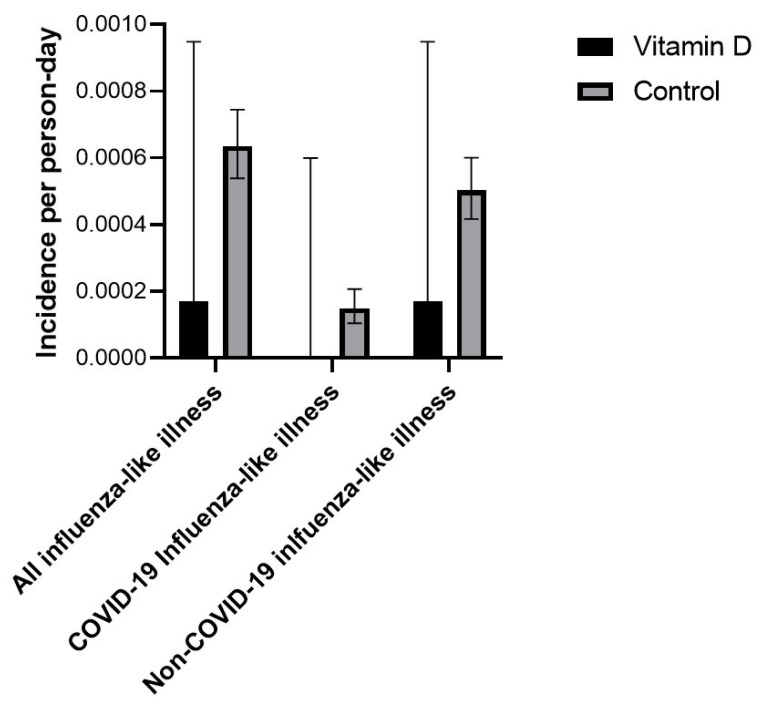
Incidence rate of influenza-like illness, COVID-19, and non-COVID-19 influenza-like illness in the vitamin D supplementation and control groups during COVID-19-non-vaccinated observation time. 95% confidence intervals are based on the methods described by Sahai and Khurshid [41].

**Figure 4 nutrients-15-00180-f004:**
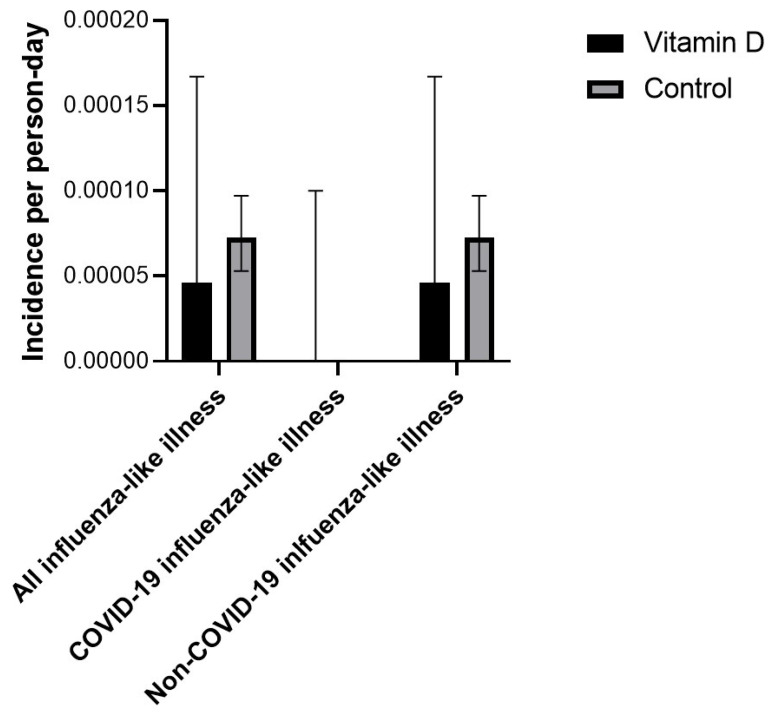
Incidence rate of influenza-like illness, COVID-19, and non-COVID-19 influenza-like illness in the vitamin D supplementation and control groups during COVID-19-vaccinated observation time. 95% confidence intervals are based on the methods described by Sahai and Khurshid [41]. The upper end of the confidence interval is not visible in the figure because it is very close to zero.

**Table 1 nutrients-15-00180-t001:** Exclusion criteria.

History of hypercalcemia History of nephrolithiasis History of intolerance to vitamin D3 supplements Use of calcium at a dose > 600 mg/day (individuals using a dose greater than 600 mg of calcium per day were asked to limit the amount to 600 mg unless they had been directed by their physician to take more than 600 mg/day. If the latter was true the potential subject was excluded from the study.) Use of vitamin D at a daily dose > 5000 IU * Use of aluminum-containing phosphate binders in patients with renal failure Use of calcipotriene Use of digoxin Use of thiazide diuretics if using: - hydrochlorothiazide at a daily dose > 37.5 mg - indapamide at a daily dose > 1.25 mg - chlorthalidone at a daily dose > 12.5 mg - metolazone at a daily dose > 2.5 mg - methyclothiazide at a daily dose > 2.5 mg - chlorothiazide at a daily dose > 250 mg - metolazone at a daily dose > 0.5 mg - bendroflumethiazide at a daily dose > 2.5 mg - polythiazide at a daily dose > 1 mg - hydroflumethiazide at a daily dose > 25 mg Conditions that are associated with a risk of modified vitamin D metabolism Known allergy to wool Current enrollment in another study Life expectancy < 1 month at time of screening Inability to provide informed consent Pregnant or trying to become pregnant Employee is team member on the present study

* If potential participants were found to be using vitamin D supplementation upon screening at a daily dose ≤ 5000 IU/day, they were eligible for participation by switching to the study dose. If potential participants were taking a multiple vitamin or calcium supplement and there was less than or equal to 800 IU vitamin D in it, they could continue the multivitamin or calcium supplement along with taking the study vitamin D3. Total vitamin D could not exceed 5800 IU per day combined between any supplements that contained vitamin D. Use of vitamin D at a daily dose > 5000 IU at the direction of a physician was an exclusion criterion. If a potential subject used over-the-counter vitamin D not directed by a physician at a daily dose > 5000 IU, they were eligible to participate by switching to the lower study dose.

**Table 2 nutrients-15-00180-t002:** Demographic and clinical characteristics of the vitamin D supplementation and control groups.

	Vitamin D3 (*n* = 255)	Control (*n* = 578)	Standardized Difference
Age at enrollment in years, mean ± SD	47 ± 12	50 ± 13	0.24
Gender, *n* (%)			
Man	55 (22)	131 (23)	0.12
Woman	199 (78)	446 (77)	0.02
Other	1 (0.4)	1 (0.2)	0.45
Race, *n* (%)			
American Indian/Alaska Native	1 (0.4)	1 (0.2)	0.45
Asian	12 (5)	36 (6)	0.04
Black/African American	27 (11)	47 (8)	0.10
Native Hawaiian/other Pacific Islander	2 (0.8)	0 (0)	0.42
White	194 (76)	456 (79)	0.07
More than one race	7 (3)	20 (4)	0.05
Other	12 (5)	18 (3)	0.10
Ethnicity, *n* (%)			
Hispanic or Latino	24 (9)	28 (5)	0.16
Not Hispanic or Latino	231 (91)	547 (95)	0.16
Body mass index in kg/m^2^, mean ± SD	30 ± 6	29 ± 6	0.17
Comorbidities, *n* (%)			
Cardiovascular disease	60 (24)	157 (27)	0.07
Respiratory disease	47 (18)	106 (18)	0
Eye disease	11 (4)	16 (3)	0.05
Gastrointestinal disease	97 (38)	213 (37)	0.02
Urological disease	19 (8)	62 (11)	0.02
Liver disease	3 (1)	8 (1)	0.10
Hematological disease	27 (11)	51 (9)	0
Dermatological disease	42 (17)	76 (13)	0.07
Diabetes	14 (6)	44 (8)	0.11
Endocrine disease (other)	38 (15)	77 (13)	0.08
Malignant disease	15 (6)	33 (6)	0
History of vitamin D deficiency, *n* (%)	62 (24)	174 (30)	0.14
Previous COVID-19, *n* (%)	16 (6)	29 (5)	0.06

**Table 3 nutrients-15-00180-t003:** Primary and secondary outcomes.

Illness	Intervention Group	Control Group	Absolute Incidence Rate Difference	95%-CI	*p*-Value	Relative Incidence Rate	95%-CI	*p*-Value
	Incidence Rate	Incidence Rate						
**All ILI**	6.104 × 10^−5^	2.288 × 10^−4^	−1.677 × 10^−4^	−3.025 × 10^−4^ to −3.3 × 10^−5^	0.0147	2.668 × 10^−1^	5.456 × 10^−2^ to 7.913 × 10^−1^	0.0060
**COVID-19 ILI**	0.0	4.181 × 10^−5^	−4.181 × 10^−5^	−9.897 × 10^−5^ to 1.536 × 10^−5^	0.1517	N/A	N/A	N/A
**Non-COVID-19 ILI**	6.104 × 10^−5^	1.916 × 10^−4^	−1.306 × 10^−4^	−2.541 × 10^−4^ to −7.1 × 10^−6^	0.0382	3.186 × 10^−1^	6.503 × 10^−2^ to 9.474 × 10^−1^	0.0229

Incidence rates are per person-day and relative incidence rate uses control group as reference; CI—Confidence Interval; ILI—Influenza-Like Illness.

**Table 4 nutrients-15-00180-t004:** Outcomes by vaccination status observation period.

Not Fully Vaccinated Observation Period	Intervention Group	Control Group	Absolute Incidence Rate Difference	95%-CI	*p*-Value	Relative Incidence Rate	95%-CI	*p*-Value
	Incidence Rate	Incidence Rate						
**All ILI**	1.701 × 10^−4^	6.354 × 10^−4^	−4.653 × 10^−4^	−1.1118 × 10^−3^ to 1.811 × 10^−4^	0.1583	2.677 × 10^−1^	6.734 × 10^−3^ to 1.5139 × 10^0^	0.1403
**COVID-19 ILI**	0.0	1.495 × 10^−4^	−1.495 × 10^−4^	−4.621 × 10^−4^ to 1.63 × 10^−4^	0.3485	N/A	N/A	N/A
**Non-COVID-19 ILI**	1.701 × 10^−4^	5.025 × 10^−4^	−3.324 × 10^−4^	−9.078 × 10−^4^ to 2.43 × 10^−4^	0.2575	3.385 × 10^−1^	8.500 × 10^−3^ to 1.9219 × 10^0^	0.2624
**Fully vaccinated observation period**	Intervention Group	Control Group	Absolute Incidence Rate Difference	95%-CI	*p*-value	Relative Incidence Rate	95%-CI	*p*-value
	Incidence Rate	Incidence Rate						
**All ILI**	4.622 × 10^−5^	7.254 × 10^−5^	−2.632 × 10^−5^	−1.0833 × 10^−4^ to 5.57 × 10^−5^	0.5294	6.372 × 10^−1^	7.488 × 10^−2^ to 2.4395 × 10^0^	0.5809
**COVID-19 ILI**	0.0	0.0	0.0	0.0 to 1.0 × 10^−4^	-	N/A	N/A	N/A
**Non-COVID-19 ILI**	4.622 × 10^−5^	7.254 × 10^−5^	−2.632 × 10^−5^	−1.0833 × 10^−4^ to 5.57 × 10^−5^	0.5294	6.372 × 10^−1^	7.488 × 10^−2^ to 2.4395 × 10^0^	0.5809

Incidence rates are per person-day and relative incidence rate uses control group as reference; CI—Confidence Interval; ILI—Influenza-Like Illness.

## Data Availability

The data presented in this study are available on request from the corresponding author. The data are not publicly available due to privacy concerns.

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
