# Peer review of "Vitamin D3 Supplementation at 5000 IU Daily for the Prevention of Influenza-like Illness in Healthcare Workers: A Pragmatic Randomized Clinical Trial"

_nutrients, 2022, doi:10.3390/nu15010180_

Round 1
Reviewer 1 Report
I was invited to revise the paper entitled "Vitamin D3 Supplementation at 5000 IU Daily for the Prevention of Influenza-Like Illness in Healthcare Workers: A Randomized Clinical Trial". It was a RCT aimed to evaluate the effectet of dailt D3 Vitamin supplementation can help in preventing ILI among HCWs. The topic is ineresting and results are in line with recent literature. Despite that, I have some major observations:
- The study was designed to evaluate the impact of supplementation on ILI but Authors did not properly specified how sample size was estimated;
- Authors defined this study as a RCT: control group was followed during the study period or not? if controls were only surveyed one time, this was not an RCT!!
- Statistical analysis was poor. Authors should perform a survival analysis. If controls were not followed during the study period, a regression analysis should be performed (Poisson? negative binomial?). Authors presented results as incidence rate difference: how was it performed? the statistical analysis was poorly condicted;
- An other limitation of this study was the lack in D Vitamin dosage among patients. We are unknown if there is a baseline difference in D Vitamin dosage between study groups. Previous literature clearly explaines that low dosage of D Vitamine in the real risk factor. The supplementation among patients with normal D vitamine value probably will not influence the study outcome;
- History of prior infection should be considered and these patients should be excluded from the analysis;
- A sub analysis should be performed: All enrolled patients should be followed till the first dose of vaccination. Vaccine is the most protective agent against covid19; In addition, differences in vaccination status should be showed;
- Authors did not consider flu vaccination. This is one more strong limitation of this study because it can influence also the ILI incidence;
- Authors decided to include all patients aged over 18. It is known that the d vitamin levels decrease across age so Authors should perform a regression model taking into account age and all comorbidities that influence D vitamin serum level (obesity, diabetes, renal diseases etc);
- Among limitations, this is not a blind trial;
- Patients enrolled in the intervention arm should be followed after a loading period of supplementation, in order to achieve a normal serum level of D vitamin. This poi in addition highlight the need to obtain the serum dosage.
Reviewer 2 Report
Thank you for the opportunity to review this manuscript. This is a a prospective, controlled trial and the aim of its was to to assess the hypothesis that vitamin D3 supplementation at 5000 IU daily reduces influenza-like illness, including COVID-19, in healthcare workers. Vitamin D or calciferol is an immunomodulatory substance that is obtained generally from exposure to sunlight and some foods. Deficiency in vitamin D has been associated with decreased lung function. Besides, deficiency in vitamin D is also reported to increase the risk of respiratory infection with Influenza A.
Generally the study is interesting and the methodology is correct buta I have some suggestions to improve the manuscript.
The introduction could be expandend with additional information. In particular I think that can be useful to mention other substance with antiviral properties like vitamin A with reference "Sinopoli A, Caminada S, Isonne C, Santoro MM, Baccolini V. What Are the Effects of Vitamin A Oral Supplementation in the Prevention and Management of Viral Infections? A Systematic Review of Randomized Clinical Trials. Nutrients. 2022 Oct 1;14(19):4081. doi: 10.3390/nu14194081. PMID: 36235733; PMCID: PMC9572963, vitamin C "Hoang, B. X., Shaw, G., Fang, W., & Han, B. (2020). Possible application of high-dose vitamin C in the prevention and therapy of coronavirus infection. Journal of global antimicrobial resistance, 23, 256-262". With these additional informations the choice of vitamin D should be better motivated from authors.
Methodology and conclusion are well written.
Round 2
Reviewer 1 Report
I was invited to review the revised version of the paper entitled "Vitamin D3 Supplementation at 5000 IU Daily for the Prevention of Influenza-Like Illness in Healthcare Workers: A Randomized Clinical Trial".
- About sample size, Authors reported "Our calculated sample size for a power of 85% was based on previous influenza-like illness occurrences in our healthcare workers. This resulted in a sample size requirement of 859 subjects". Authors stated only the power level, without presenting the exact calculation performed;
- About the definition of RCT, Authors should state in the title and in methods that this is a pragmatic trial;
- Statistical analysis description was not improved;
- In response letter, Authors stated "If baseline vitamin D level was influential, we believe it was non-differential on the control and intervention groups given the similar demographics and comorbidities of the two groups". In reality, in table 2 several characteristics present a standardized mean difference over 10% so there is a difference in baseline characteristics between study groups;
- Authors did not presented adequate responds to flu vaccination questions;
